# Innovating the Synergistic Assets of β-Amino Butyric Acid (BABA) and Selenium Nanoparticles (SeNPs) in Improving the Growth, Nitrogen Metabolism, Biological Activities, and Nutritive Value of *Medicago interexta* Sprouts

**DOI:** 10.3390/plants11030306

**Published:** 2022-01-24

**Authors:** Samy Selim, Nosheen Akhtar, Eman El Azab, Mona Warrad, Hassan H. Alhassan, Mohamed Abdel-Mawgoud, Soad K. Al Jaouni, Hamada Abdelgawad

**Affiliations:** 1Department of Clinical Laboratory Sciences, College of Applied Medical Sciences, Jouf University, Sakaka 72341, Saudi Arabia; h.alhasan@ju.edu.sa; 2Department of Biological Sciences, National University of Medical Sciences, Rawalpindi 46000, Pakistan; nosheenakhtar@numspak.edu.pk; 3Department of Clinical Laboratory Sciences, College of Applied Medical Sciences at Al-Quriat, Jouf University, Al-Quriat 77454, Saudi Arabia; efelazab@ju.edu.sa (E.E.A.); mfwarad@ju.edu.sa (M.W.); 4Department of Medicinal and Aromatic Plants, Desert Research Centre, Cairo 11753, Egypt; Mohamed_drc@yahoo.com; 5Hematology/Pediatric Oncology, Yousef Abdulatif Jameel Scientific Chair of Prophetic Medicine Application, Faculty of Medicine, King Abdulaziz University, Jeddah 21589, Saudi Arabia; saljaouni@kau.edu.sa; 6Botany and Microbiology Department, Faculty of Science, Beni-Suef University, Beni-Suef 62511, Egypt

**Keywords:** *Medicago interexta*, sprouts, SeNPs, BABA, nutritious metabolites, anti-diabetic

## Abstract

In view of the wide traditional uses of legume sprouts, several strategies have been approved to improve their growth, bioactivity, and nutritive values. In this regard, the present study aimed at investigating how priming with selenium nanoparticles (SeNPs, 25 mg L^−1^) enhanced the effects of β-amino butyric acid (BABA, 30 mM) on the growth, physiology, nitrogen metabolism, and bioactive metabolites of *Medicago interexta* sprouts. The results have shown that the growth and photosynthesis of *M. interexta* sprouts were enhanced by the treatment with BABA or SeNPs, being higher under combined treatment. Increased photosynthesis provided the precursors for the biosynthesis of primary and secondary metabolites. In this regard, the combined treatment had a more pronounced effect on the bioactive primary metabolites (essential amino acids), secondary metabolites (phenolics, GSH, and ASC), and mineral profiles of the investigated sprouts than that of sole treatments. Increased amino acids were accompanied by increased nitrogen metabolism, i.e., nitrate reductase, glutamate dehydrogenase (GDH), glutamate synthase (GOGAT), glutamine synthase (GS), cysteine synthesis serine acetyltransferase, arginase, threonine synthase, and methionine synthase. Further, the antioxidant capacity (FRAP), the anti-diabetic activities (i.e., α-amylase and α-glucosidase inhibition activities), and the glycemic index of the tested sprouts were more significantly improved by the combined treatment with BABA and SeNPs than by individual treatment. Overall, the combined effect of BABA and SeNPs could be preferable to their individual effects on plant growth and bioactive metabolites.

## 1. Introduction

Phytochemicals are important metabolic compounds that confer the capability of plants to combat environmental stress, boost their defense systems, and protect them from pathogens and insects. Secondary metabolites also play key roles as health-promoting enzymes; accordingly, they are an essential part of human health. These metabolites, especially the phenolic compounds [1] and glucosinolates [2], have been reported for their protective effects against the oxidative process and provide protection against different diseases, such as cardiovascular diseases, neurodegenerative diseases, and cancer [3]. Currently, the development of new strategies to improve plant growth and boost the production of secondary metabolites is one of the fascinating fields of research. Application of elicitors and nanoparticles can enhance the production of bioactive metabolites in plants, including their qualitative value in producing fresh produce, enriched foods, or raw ingredients for feed/food and pharmaceutical products [4,5].

Elicitors mimic the action of plant signaling and increase the production of reactive oxygen species (ROS) and reactive nitrogen species (RNS), upregulate the defense-related genes, change the potential of plasma membrane cells, and enhance ion fluxes (Cl^−^ and K^+^ efflux and Ca^2+^ influxes) [6,7]. They also induce changes in protein phosphorylation and lipid oxidation, and activate the de novo biosynthesis of transcription factors, which directly regulate the expression of genes involved in secondary metabolites production [6,7]. β-aminobutyric acid (BABA) is a nonprotein amino acid that is considered as one of the plant activators that induce resistance in many different plant species against a wide range of abiotic and biotic stresses. BABA, which was found to present naturally at low concentrations in plant tissues, can be increased 5-fold to 10-fold under stress conditions [8]. The broad spectrum protective effect of BABA against numerous plant diseases has been well documented [9,10] and is attributable to enhanced phenolics content or related compounds. For example, research has shown that BABA induces changes in the response of leaf antioxidants to UV-B [11,12]. Moreover, BABA interacts with several hormones, such as salicylic acid (SA), abscisic acid (ABA), and ethylene [8] and thereby takes part in the growth of plants, including development, photosynthesis, transpiration, and ion uptake and transport.

Furthermore, in the context to plant growth, nanoparticles have unique physicochemical properties and the potential to boost plant metabolism, and thus the production of secondary metabolites [13]. Application of nanoparticles (NPs) is currently an interesting area for minimizing the use of chemical fertilizers and improving the growth and yield of plants [14,15]. The unique physicochemical properties of NPs have potentially opened up new paradigms, and the introduction of NPs to plants might have a significant impact; therefore, they can be used in agricultural applications for better growth and yield. 

Among different nanoparticles, selenium nanoparticles (SeNPs) have precedence over other nanoparticles because of the significant role of selenium in activating plants’ defense systems. Several studies have demonstrated that Se may exert diverse beneficial effects at low concentrations as an antioxidant and as a growth-promoting agent in higher plants. Moreover, some plants are able to accumulate large amounts of Se as an essential element [16]. 

Se uptake by plants depends on some environmental factors, such as soil pH, salinity, and concentration of competing ions. Usually, the stems and leaves accumulate higher Se levels than do the roots [17]. It has also been demonstrated that Se might affect plant growth and many metabolic processes. For instance, Se might contribute to maintaining the water potential of plants under drought conditions [18]. Se could enhance light harvesting, thereby increasing the available energy for plants [17]. On the other hand, the phytoxicity of Se might be related to an interaction with sulfur; consequently, sulfur-containing amino acids might be replaced by Se-containing amino acids [19].

The toxicity of Se depends on its chemical form as well as on plant age. Se toxicity could be observed at a concentration of ≥2 mg/kg dry weight. The maximum Se content (safest concentration) in the medium without growth inhibition was found to be 1, 10, 0.25, and 0.25 mg/L for radish, sunflower, alfalfa, beetroot, respectively [20]. On the other hand, SeNPs have a more enhancing effect on plants, with low toxicity, when compared with the bulk form [21]. In addition, the use of biogenic SeNPs is known to be an environmentally friendly and ecologically biocompatible approach in enhancing crop production by alleviating biotic and abiotic stresses [22]. Moreover, SeNPs enhance photosynthetic pigment activity, nutrient status, antioxidant activity, and total phenolic content under drought stress. Surprisingly, at a minimal dose, Se is highly effective against salinity stress by maintaining turgor pressure, controlling the accumulation of total sugars, amino acids, and potential antioxidant enzymes, and improving the transpiration rate [22]. Se also decreases chloride ion contents, ROS species, and membrane damage. In addition, Se decreases sodium-ion accumulation and increases potassium-ion accumulation, thereby reducing the detrimental effects of salt stress on plants [23].

Legumes are valued worldwide as a sustainable and inexpensive meat alternative and are considered the second most important food source after cereals. Legumes are a rich source of many nutrient components, including starch, protein, certain fatty acids, and micronutrients such as vitamins, minerals, and bioactive compounds [24,25,26]. Medicago is the genius of leguminous plants and *Medicago interexta* (*M. interexta*) is an important member, reported to be the source of proteins and tannins [26]. Regarding the significance of BABA and SeNPs in triggering the production of phytochemicals, we hypothesized that the application of both can have additive effects and can enhance the nutritional and pharmacological value of *M. interexta* by improving the production of primary and secondary metabolites. Thus, the present study aimed to evaluate the impact of BABA, SeNPs, and their combined effects on *M. interexta* sprouts. We evaluated the impacts on growth, mineral content, the vitamin and amino acid profile, nitrogen, and phenolic metabolism, as well as on the concentrations of several phytochemical compounds. We further examined the role of SeNPs and/or BABA in the enhancement of the antioxidant and antidiabetic potential of *M. interexta*. Overall, our study contributes to an understanding of the biochemical basis of BABA, SeNPs, and their combination in *M. interexta*.

## 2. Results

### 2.1. Enhanced Growth of M. interexta Sprouts under Sole and Combined Treatments with BABA and/or SeNPs

The present investigation revealed that the treatment of *M. interexta* with β-amino butyric acid (BABA) led to a significant increase in biomass accumulation (expressed as fresh weight FW, dry weight DW), photosynthesis, and respiration by approximately 40%, in comparison to control sprouts (Figure 1). The addition of SeNPs to the target sprouts also induced a higher increase in growth and photosynthesis of *M. interexta* sprouts (by about 50–90%), in comparison to the non-treated plants. Interestingly, the combined effect of BABA and SeNPs resulted in a much higher increment in growth parameters, by approximately 200% when compared with the control sprouts. Thus, the growth of *M. interexta* sprouts was enhanced by the sole and combined treatment with BABA and/or SeNPs, with higher enhancement under the combined treatment. 

Regarding pigment content, the sole treatment of *M. interexta* sprouts with BABA significantly increased almost all carotenoids (i.e., chl a, b, β-carotene, lutein, neoxanthin, and violaxanthin) (Table 1). In addition, when *M. interexta* sprouts were grown under individual treatment with SeNPs, there were significant increments in all the detected carotenoids, except for neoxanthin. Moreover, the combined treatment of *M. interexta* sprouts with BABA and SeNPs increased all the detected carotenoids, when compared with the control sprouts.

### 2.2. Combined Treatment of M. interexta Sprouts with BABA and SeNPs Induced a More Pronounced Effect on Mineral and Vitamin Profiles than That of a Sole Treatment

In the current study, the mineral and vitamin profiles were investigated in *M. interexta* under the different effects of BABA and/or SeNPs (Table 2). Under control conditions, eight mineral elements (Ca, Cu, Fe, Zn, Mn, Mg, K, and P) were detected in *M. interexta* sprouts, whereas Zn had the highest content, followed by Ca and K. When *M. interexta* sprouts were treated individually with BABA, there was a significant increase only in Zn (by about 70%), in addition to a significant decrease in Mn, while no changes were observed for Cu, Fe, Ca, or K. In the case of sole treatment of *M. interexta* sprouts with SeNPs, there were remarkable increases in Ca, Fe, Zn (increased by 60–80%), K, and P (increased by approximately 100–150%), while no significant changes were reported for Cu and Mn. On the other hand, the combined treatment of *M. interexta* sprouts with BABA and SeNPs induced enhancing effects on the contents of Ca (elevated by 50%), Fe, Zn, Cu (increased by 80–100%), K (450%), and P (increased by about 170%). It was observed that Mn was not affected by any of the treatments used.

Regarding vitamin content, four vitamins (Vit C, Vit E, thiamin, and riboflavin) were detected in *M. interexta* sprouts under control conditions, wherein Vit E was the predominant vitamin (Table 2). When treated individually with BABA, the target sprouts did not show significant changes in vitamin content, except for riboflavin (increased by 80%), in comparison to control plants. Similarly, there were no significant differences in vitamins, except for thiamin, in response to the sole treatment with SeNPs. Meanwhile, the interactive impact of both BABA and SeNPs has been reflected on increasing all vitamins detected in comparison to the control. Overall, the combined treatment with BABA and SeNPs had a more pronounced effect on the mineral and vitamin profiles of *M. interexta* sprouts than did a sole treatment.

### 2.3. M. interexta Sprouts Were More Responsive to the Combined Effect of BABA and SeNPs on Nitrogen Metabolism than to Individual Treatments

In the present investigation, amino acids have been analyzed in *M. interexta* sprouts grown under higher concentrations of BABA and/or SeNPs (Table 3). Under control conditions, 18 amino acids (i.e., asparagine, glutamine, serine, glycine, arginine, alanine, proline, histidine, valine, methionine, cystine, ornithine, leucine, phenylalanine, tyrosine, lysine, threonine, and tryptophane) were quantified in *M. interexta*, where glutamine had the highest percentage. From the current data, it is clear that *M. interexta* sprouts interacted differently to the effects of BABA and/or SeNPs. There were significant elevations in the contents of serine, glycine, alanine, proline, histidine, valine, ornithine, and phenylalanine, while no significant changes were observed for asparagine, glutamine, cystine, leucine, arginine, methionine, lysine, threonine, tryptophane, or tyrosine in *M. interexta* sprouts treated solely with BABA, when compared with the control sprouts. 

The individual treatment of *M. interexta* sprouts with SeNPs markedly induced the contents of asparagine, glutamine, serine, arginine, proline, methionine, cystine, ornithine, tyrosine, threonine, and tryptophane, but there were no significant changes in the levels of glycine, alanine, histidine, valine, leucine, lysine, or phenylalanine, when compared with the control sprouts. Moreover, the interaction between BABA and SeNPs led to significant increments in most of the detected amino acids in *M. interexta* sprouts, except for histidine, valine, and leucine, when compared with control plants.

Regarding nitrogen metabolism, *M. interexta* sprouts interacted differently to the effects of BABA and/or SeNPs on N, total protein, nitrate reductase, GDH, GOGAT, GS, cysteine synthesis serine acetyltransferase, arginase, threonine synthase, and methionine synthase (Table 4). When *M. interexta* sprouts were grown under the individual impact of BABA, there were remarkable increases in N content, nitrate reductase, GDH, GOGAT, GS, cysteine synthesis serine acetyltransferase, arginase, threonine synthase and methionine synthase, as well as significant reductions in total protein, in comparison to the control sprouts. In the case of treatment individually with SeNPs, the tested sprouts tended to display notable increases in N, GOGAT, GS, cysteine synthesis serine acetyltransferase, arginase, and methionine synthase, in addition to significant decreases in total protein, while no changes were reported for nitrate reductase, GDH, or threonine synthase, when compared to control sprouts. On the other hand, the combined treatment of *M. interexta* sprouts with BABA and SeNPs positively influenced the levels of all the measured related N-parameters, except for total proteins, which were significantly decreased when compared with control sprouts. It could be noted that the interaction between BABA and SeNPs exerted a more pronounced effect on the nitrogen metabolism of *M. interexta* than their individual treatments.

### 2.4. Antioxidants of M. interexta Sprouts Were Improved by the Sole and Combined Treatments with BABA and/or SeNPs

The levels of antioxidants (i.e., phenolics, FRAP, CAT, POX, GSH, and ASC) were measured in the target sprouts under the impact of BABA and/or SeNPs (Table 5). The individual treatment of *M. interexta* sprouts with BABA resulted in significant increases in flavonoids, phenols, FRAP (by about 90%), and GSH (by about 20%), as well as in ASC content (by about 80%), in comparison to control plants. Meanwhile, the sole treatment of the target sprouts with SeNPs also increased the levels of flavonoids, phenolics, antioxidant activity (by about 90%), GSH (by about 20%), and ASC content (by 20%), when compared with the control sprouts. Interestingly, highly significant increases in flavonoids, phenols, FRAP (by about 130%), GSH, and ASC (by about 100%) were obtained in *M. interexta* sprouts when treated with the combination of BABA and SeNPs. Thus, the levels of antioxidants of *M. interexta* were enhanced by the sole and combined treatments with BABA and/or Se NPs, with higher enhancement under the combined treatment.

### 2.5. Anti-Diabetic Activity of M. interexta Sprouts Was More Improved by the Combined Treatment with BABA and Se NPs than by Individual Treatments

In the present study, anti-diabetic activity (i.e., α-amylase and α-glucosidase inhibition activities, and the glycemic index GI) was investigated in *M. interexta* sprouts in response to the different effects of BABA and/or SeNPs (Figure 2). When treated individually with BABA, *M. interexta* sprouts showed more increases in α-amylase and α-glucosidase inhibition activity (by about 40% and 20%, respectively), as well as a significant decrease in GI (by about 50%) in comparison to the control. Meanwhile, the sole treatment of *M. interexta* sprouts with SeNPs induced significant increments in both α-amylase and α-glucosidase inhibition activities (increased by 20% and 10% respectively), but it decreased the GI (by about 30%). Interestingly, the interactive impact imposed by BABA and SeNPs has induced the levels of α-amylase and α-glucosidase inhibition activities, (by about 50%, and 90%, respectively), but decreased the GI (by about 30%), when compared with control plants. Thus, the anti-diabetic activity of *M. interexta* sprouts was more improved by the combined treatment with BABA and SeNPs than by an individual treatment.

## 3. Discussion

The present study was conducted to explore the collective effects of BABA and SeNPs on *M. interexta* sprouts in enhancing resistance against infections and increasing nutritional and pharmacological values. The effects of BABA and SeNPs on biosynthetic pathways and on the biological activities of *M. interexta* sprouts were evaluated, both alone and in combination. SeNPs and BABA have emerged as part of an effective class of elicitors that induce a defense mechanism that enhances the production of valuable bioactive metabolites. Our results indicated that the intervention comprised of a combined BABA and SeNPs treatment had a more significant impact on the endogenous biosynthetic pathways of *M. interexta* sprouts, as compared to individual treatments.

### 3.1. Improved Growth of M. Interexta Sprouts

In the current study, significant increases in the biomass production and photosynthetic activity of *M. interexta* sprouts were observed under treatment with BABA and SeNPs alone; however, the increases were remarkable when both agents were used in combination. The increases might be attributed to the additive effects of BABA and SeNPs that elicited a vigorous increase of metabolism and mineral content, as measured in our study. Many previous studies described the growth-modulating effects of BABA on different plants. Jisha et al. reported that BABA seed-priming increased seedling growth, under both unstressed and stressed conditions in rice [27]. BABA has been thought to enhance nitrogen metabolism, which consequently provides precursors needed for the biosynthesis of amino acids and protein and increases photosynthesis, growth rates, and biomass accumulation. In addition, the improved photosynthetic pigments under treatment with BABA, as reported herein, are directly related to the photosynthesis process and to the efficiency of photosynthesis. We observed that the increase of biomass using SeNPs was higher, as compared to using BABA. However, a remarkable increase in plant growth was observed when a combined treatment used both BABA and SeNPs. The positive effects of SeNPs on the growth of different plants support our data on the increased growth of *M. interexta* when SeNPs were used, either alone or in combination. 

Previous studies showed that the use of SeNPs indicated growth-promoting effects in cowpea yield [28], efficiently upregulated selenoenzymes, and exhibited less toxicity [29]. Previous studies have also shown that SeNPs could enhance the photosynthetic efficiency of some plants, such as tomato. Such positive effects could also be reflected in increasing pigment contents, as reported in our study. This might be due to the small size of NPs, enabling them to easily move through plant parts [30]. In tomato, SeNPs improved the parameters of plant growth at low concentration (1 μM) [31]. Similarly, SeNPs at 400 mg improved the growth of the cluster bean [32].

### 3.2. Improved Pigment Content of M. interexta Sprouts

Interesting patterns were observed in the pigment contents of sprouts, using individual and combined treatment groups of *M. interexta*. The differential patterns indicated that the combined treatment targeted multiple pathways that were not affected when a single agent was used. For example, the combined treatment increased Chl a, Chl b, and neoxanthin, while BABA alone also increased these pigments. At a concentration of 25 mg L^−1^, SeNPs’ suspension-priming significantly reduced neoxanthin when used alone, while in sprouts subjected to combined treatment, neoxanthin was observed to be increased. Similarly, the combined intervention and the sole treatment with SeNPs or BABA resulted in significant increases in violaxanthin.

Previous studies reported the effects of SeNPS and BABA on photosynthetic pigments. SeNPs at a low concentration of 6.25 μM were found to be effective in increasing total photosynthetic pigments in the leaves of cowpea [28]. Similarly, in tomato leaves, application of SeNPs at 1 μM improved the chlorophyll content by 27.5% [31]. Contrary to our results, the priming of seeds with BABA is reported to have positive effects on pigment content. For example, rice seed-priming with BABA increased the photosynthetic pigment content of leaves, modified the Chl a fluorescence, and enhanced the photosystem activities of seedlings [27].

Our study results are also contrary to the reported finding that BABA exhibited an undesirable side effect, i.e., that it reduces plant growth [33]; however, we observed that BABA alone also enhanced photosynthesis and plant growth. This is attributable to the fact that different plant species employ different defense mechanisms and, accordingly, differential effects of the same elicitor can be observed among species. Our results showed that the combined treatment of *M. interexta* could increase the content of Chl a, Chl b, and carotene significantly, indicating that it could strengthen MI by enhancing the photosynthetic system.

### 3.3. Improved Mineral Content and Vitamin Profile of M. interexta Sprouts

Plant-derived foods have the potential to serve as dietary sources for all human-essential minerals. The essential minerals include N, S, P, K, Ca, Cl, Fe, Zn, Mn, Cu, B, Mo, and Ni. Among these, Ca, Zn, Ca, Cu, Fe, K, Mn, K, and P were detected in *M. interext* sprouts, from which Zn, Ca, and K were present in higher amounts. We evaluated the effects of treatments on mineral content and the results revealed that Zn concentration was increased by BABA while SeNPs increased K and P. The combined treatment resulted in a robust increase in the concentration of K, P, Fe, Zn Cu, and Ca. These increases in minerals might lead to remarkable increases in the growth of sprouts of *M. interexta*, as the minerals, especially K, modulate various biochemical and physiological processes that are responsible for plant growth and development. Also, the BABA-induced increases in minerals could be due to increased root growth that, in turn, triggers nutrient uptake by plants [34,35]. BABA upregulated mineral transporters [36,37]. Moreover, improved nitrogen nutrition by BABA treatment could enhance root uptake, root-to-shoot translocation, and remobilization of Zn [38]. In this context, the positive effects of nitrogen and Zn uptake and translocation can be explained by upregulating the transporter proteins and nitrogenous chelators involved in these processes. Consequentially, an increased level of Zn is needed for biosynthesis and for the structural and functional integrity of proteins and amino acid metabolism [39]. 

Regarding the effect of SeNPs on mineral uptake, the study in [40] indicated that exposure to Se significantly upregulated the expressions of the phosphate transporter (PHT), the potassium channel protein (KCP), and the potassium transporter protein (KTP). In agreement with our results, Se application was found to enhance the mineral content (e.g., Zn, Mn, Cu, Ca, Mg, Na, and K) of alfalfa and radish [20]. It was found that the mineral content (P, K, Ca, and Mg) of garlic was significantly reduced under Se treatment [41]. Furthermore, our results showed that *M. interexta* sprouts are a rich source of vitamins, especially vitamin E, which were further increased by the combined treatment with BABA and SeNPs. High N availability under BABA treatment can also promote plants’ Se absorption, and Se can then be further metabolized into seleno-proteins. In this regard, N fertilizer promotes growth, thereby promoting the absorption of P, K, S, and other mineral elements, including Se, by the root system [42].

Interestingly, neither BABA nor SeNPs had an effect on the concentrations of vitamins when used alone. Our study indicated that mutual intervention was more effective in triggering the multiple defense pathways that consequently enhanced the concentration of vitamins.

### 3.4. Improved Nitrogen Metabolism of M. interexta Sprouts

It is known that the nitrogen source, either nitrate or ammonium, affects the levels of amino acids and proteins, and consequently the rate of growth and biomass accumulation. Nitrogen metabolism is thought to be involved in the conversion of amino acids via nitrate reduction [43]. BABA is thought to enhance nitrogen metabolism, which consequently provides precursors that are needed for the biosynthesis of amino acids and protein. Previous reports have also shown that priming could increase nitrogen metabolism by enhancing the contents of amino acids and total protein, as well as nitrate reductase activity [44].

In our study, the individual and combined treatments with BABA and/or SeNPs have positively affected almost all the measured N-related parameters. In line with our results, priming has been shown to increase the production of GDH and GOGAT [44]. In this regard, the GS/GOGAT pathway is thought to assimilate ammonia at normal intracellular concentrations, while GDH plays a role in the assimilation of ammonia into amino acids. Similarly, γ-aminobutyric acid (GABA) has been previously found to promote total nitrate reductase activity [45].

Arginase is known to be involved in the conversion of arginine into ornithine, so it might contribute to increasing the ornithine content in the sprouts treated by BABA and/or SeNPs, as reported in our study. Consequently, ornithine could act as a precursor for the synthesis of polyamines and some amino acids, such as glutamate and proline, which are incorporated into many physiological processes, particularly under stress conditions [46]. In addition, arginase plays a role in increasing some other amino acids by providing the carbon and nitrogen skeleton required for their biosynthesis [47]. Further, the hydrolysis of arginine by arginase results in formation of urea, which in turn is hydrolyzed into ammonia. Finally, ammonia is involved in the glutamine synthetase/glutamine oxoglutarate aminotransferase (GS/GOGAT) cycle [46].

### 3.5. Improved Antioxidants of M. interexta Sprouts

Previous studies showed that BABA enhances a variety of plant metabolites and their associated mechanisms, and thus strengthens the defense systems of plants. BABA promotes the synthesis of phenolics and anthocyanins, and elevates the production of the enzymes associated with ROS [33]. Zhong et al. reported that BABA enhanced the activation of defense enzymes in soybean [48]. BABA also has been reported to potentiate different defence-signaling pathways under biotic and abiotic stresses [49]. Similarly, biogenic SeNPs improve the antioxidant defensive system of plants under abiotic stress [50]. SeNPs were also reported to be significantly involved in quenching ROS due to enhanced production of antioxidant enzymes, including guaiacol peroxidase (GPX), superoxide dismutase (SOD), proline oxidase (POX), and catalase (CAT) [51,52]. 

In the present study, we observed that both BABA and SeNPs increased the concentrations of phenolics and flavonoids, which might be attributed to enhanced antioxidant activity, as indicated by the results of FRAP assay. However, the increase was more significant under the combined treatment. Previous studies have shown the ability of BABA and/or Se to increase the levels of phenolics in plants grown under stress conditions [53,54]. Such induced increments might be due to activation of phenylalanine ammonia-lyas (PAL), which is a key enzyme in the phenylpropanoid pathway, as it is responsible for the biosynthesis of phenolic compounds [53,54]. Similarly, phenolic compounds were previously enhanced in potato when treated with BABA [55]. In addition, the PAL content of garlic has been found to be significantly increased under Se treatment, thereby enhancing its phenolic content [54]. In addition, the induced photosynthetic activity under treatment such as BABA and/or SeNPs could significantly increase the carbon skeleton necessary for the biosynthesis of different classes of secondary metabolites, such as phenolic compounds [56,57,58]. Moreover, the remarkable rise in GSH, the key non-enzymatic antioxidant, was measured in the combined treatment. The ameliorated ratio of GSH/GSSG is required for the generation of ascorbate (ASC) and the stimulation of numerous CO_2_-fixing enzymes in the chloroplasts [59], ensuring the availability of NADP^+^ to accept electrons from the photosynthetic electron transport chain.

### 3.6. Improved Antidiabetic Activity of M. interexta Sprouts

As the pharmacological properties of plants are correlated to their phytochemical content, we explored the enhanced phytochemical content that is attributed to the enhancement of the antidiabetic potential of *M. interexta* sprouts. We evaluated α-amylase and α-glucosidase inhibition activities, and the glycemic index of *M. interexta* sprouts. Results indicated that BABA increased the α-amylase inhibition activity of MI, while the GI of *M. interexta* sprouts was significantly decreased. SeNPs had positive effects in increasing the inhibition activity against α-amylase and α-glucosidase. Notably, the combined treatment increased the inhibitory effects against both enzymes but decreased the GI. A large variety of α-amylase and α-glucosidase inhibitors have been reported from various plants [60]. The reported inhibitory enzymatic activity in our study may be due to the presence of potentially bioactive compounds, such as polyphenols, alkaloids, flavonoids, tannins, and glycosides, which can enhance the combined treatment of BABA and SeNPs, leading to an increased antidiabetic potential of *M. interexta* sprouts.

### 3.7. Species-Specific Response to BABA and/or SeNPs

To better understand the BABA- and/or SeNPs-induced effects on *M. interexta* sprouts, we performed a principal component analysis (PCA) of the chemical composition and biological activities of the tested sprouts (Figure 3). There was a clear separation between the treatment parameters along the PC1, which explains 67% of the total variation. Obviously, the combined treatment of BABA and SeNPs induced the accumulation of amino acids, vitamins, and many components, as well as antidiabetic activity. There was also a clear separation between the parameters of the individually treated BABA or SeNPs sprouts along PC2 (representing 29% of the total variation). The sole treatment of *M. interexta* sprouts with SeNPs enhanced higher amounts of amino acids, vitamins, and other components, compared with sole treatment with BABA. Overall, the present data showed that *M. interexta* sprouts were differentially grouped, indicating the specificity of the accumulation of nutritive metabolites in response to the individual and/or the combined treatments with BABA and/or SeNPs.

## 4. Materials and Methods

### 4.1. Experimental Setup

Seeds of *M. interexta* were collected from the Agricultural Research Centers, where they were collected during filed trips to different locations in Egypt (Giza and Ismailia) and Saudi Arabia (Riyadh, Saudi Arabia). Seeds of *M. interexta* were collected from Dr. Mohammad K. Okla, Botany and Microbiology Department, College of Science, King Saud University, Riyadh, Saudi Arabia. The seeds were soaked for 1 h in 5 g L^−1^ of sodium hypochlorite for disinfection, and then they were washed with distilled water. The plant seeds were divided into two groups: the first group was primed with suspension containing 25 mg L^−1^ of selenium nanoparticles (SeNPs) for 10 h with continuous shaking (shaker (IKA KS 501 shaker, Staufen, Germany) at room temperature (24 °C). Then, the seeds were washed thrice with distilled water for 2 min. For sprouting processes, the seeds of both groups (200 seeds per group) were distributed on trays (3 trays/treatment) filled with vermiculite and irrigated with 200 mL of 30 mM β-amino butyric acid (BABA) solution. The control trays were irrigated with Milli-Q water. Then, the seeds were evenly transferred to trays and covered. The applied concentrations of BABA and SeNPs were selected according to pilot experiments, where six concentrations of BABA (0 (distilled water) and 5, 15, 30, 60, and 90 mM) and 5 concentrations of SeNPs (0 (distilled water), 10, 25, 50, 75 mg L^−1^) were tested. The growth conditions were adjusted to 150 μmol (photosynthetically active radiation) PAR m^−2^ s^−1^, 23/18.5 °C air temperature, 63% humidity, and 16/8 h day/night photoperiod. Each experiment was replicated two times, and for all assays, four biological replicates (two biological replicates from each experiment) were used; accordingly, 16 samples in total were analyzed per each measurement. Moreover, each replicate corresponded to a group of 25 sprouts harvested from a certain tray. The sprout tissues (leaves and stems) from each treatment were harvested after 9 days. After fresh weight (FW) and dry weight (DW) measurements, the sprouts were frozen in liquid nitrogen and kept at −80 °C for biochemical analysis.

### 4.2. Selenium Nanoparticles Characterization

Selenium nanoparticles (SeNPs) were purchased from American Elements (Los Angeles, CA, USA) (https://www.americanelements.com/selenium-nanoparticles-7782-49-2, accessed on 25 February 2017). They are gray to black solids of a size of 20 and a specific surface area of 40 m^2^/g, purity of 99.99%, and a density of 4.79 g/cm^3^, according to the manufacturer’s data. The morphological features were validated by using a scanning electron microscope (SEM manufacturered by JEOL JSM-6510, LA, Japan). To avoid coarse aggregation of SeNPs in aqueous solution, NPs were sonicated.

### 4.3. Determination of Photosynthetic Rate

Photosynthesis (μmol CO_2_ m^−2^ s^−1^) and dark respiration (μmol CO_2_ m^−2^ s^−1^) of the treated sprouts were detected by using an EGM-4 infrared gas analyzer (PP Systems, Hitchin, UK). Photosynthesis dark respiration was determined from 180 s measurements of net CO_2_ exchange (NE)**.**

### 4.4. Pigment Analysis

For homogenization of sprout samples, a MagNALyser (Roche, Vilvoorde, Belgium) was used for 1 min at 7000 rpm, then centrifugation was done for 20 min at 4 °C and 14,000× *g*. The supernatant was filtered through an Acrodisc GHP filter (0.45 μm 13 mm) (Gelman, Ann Arbor, MI, USA) and was further analyzed by HPLC (Shimadzu SIL10-ADvp, Kyoto, Japan, reversed-phase, at 4 °C). Pigments were separated on a C18 silica column (Waters Spherisorb, 5 μm ODS1, 4.6 × 250 mm, at 40 °C), using a mobile phase, as follows: (A) 81:9:10 acetonitrile/methanol/water and solvent; (B) 68:32 methanol/ethyl acetate, at a flow rate of 1.0 mL/min at room temperature [61]. A diode-array detector (Shimadzu SPD-M10Avp, Kyoto, Japan) was used for detection of chlorophyll *a* and *b*, and β-carotene at 420, 440, and 462 nm. Shimadzu Lab Solutions Lite software was used for the calculation of concentrations.

### 4.5. Analysis of Mineral Contents

Detection of mineral elements was carried out according to [62,63], whereas 200 mg from treated and control plants grown were digested by using an HNO_3_/H_2_O solution (5:1). Thereafter, macro- and micro-elements were evaluated by using inductively coupled plasma mass spectrometry (ICP-MS, Finnigan Element XR, and Scientific, Bremen, Germany). Nitric acid (1%) was used as a standard.

### 4.6. Determination of Phenolic, Flavonoid Contents, and Vitamins Levels

To extract phenolics and flavonoids, 150 mg of sprout material were extracted in 2 mL 80% methanol. Then, it was homogenized by a MagNALyser (Roche, Vilvoorde, Belgium; 7000 rpm/1 min). The extraction was performed three times. After each extraction, samples were centrifuged at 4 °C 20 min at 10,000× *g*, then the supernatants were transferred to clean tubes. The resulting supernatants were combined and centrifuged again at 4 °C for 30 min at 10,000× *g* to remove suspended particles. Prior to analysis, the samples were diluted 1:2 in 80% methanol, and 10 μL was used. The phenolic content was determined by using a Folin–Ciocalteu assay, where gallic acid was used as a standard [58]. The flavonoid content was evaluated following the modified aluminum chloride colorimetric method, where quercetin was applied as a standard [58]. The levels of phenolic and flavonoid compounds were identified by HPLC methods using the standards and their relative retention times, whereas the peak area of each standard could be used as an indication of the amount of each compound. For detection of the target compounds, approximately 50 mg samples were mixed with acetone/water (4:1). The HPLC system (SCL-10 AVP, Kyoto, Japan) was provided with a Lichrosorb Si-60, 7 μm, 3 mm × 150 mm column and a diode array detector. The mobile phase was a mixture of (90:10) water/formic acid, as well as (85:10:5) acetonitrile/water/formic acid, at a flow rate of 0.8 mL/min. The binary solvent system utilized in the mobile phase consisted of the following: (A) 1 percent acetic acid/water, and (B) methanol, with the gradient being 0 min 40% B, 5 min 65 percent B, 10 min 90% B, and 15 min 40% B until 17 min, as modified from the reference. The eluate was tested for UV absorbance at 260, 280, and 330 nm. Compounds were found by comparing retention times, absorbance spectrum profiles, and running samples, after pure standards had been added to known concentrations of each discovered compound to internal standards. Meanwhile, the internal standard was 3,5-dichloro-4- hydroxybenzoic.

Detection of vitamins in treated and control sprouts was carried out via HPLC, according to [58,64]. The contents of thiamine and riboflavin were determined in sprouts, by using UV and/or fluorescence detectors [58]. Separation was performed on a reverse-phase (C18) column (HPLC, methanol/water). Ascorbate (Vit C) was extracted in 1 mL of 6% (*w*/*v*) meta-phosphoric acid at 4 °C and was separated by reverse-phase HPLC coupled with a UV detector (100 mm × 4.6 mm Polaris C18-A, 3 lm particle size; 40 °C, isocratic flow rate: 1 mL min^−1^, elution buffer: 2 mM KCl, pH 2.5 with O-phosphoric acid). Tocopherol (vit E) was separated on Particil Pac 5 µm column material (length 250 mm, i.d. 4.6 mm) and quantified by HPLC (Shimadzu’s Hertogenbosch, s-Hertogenbosch, The Netherlands, normal phase conditions), coupled with a fluorometric detector (excitation at 290 nm and emission at 330 nm). Riboflavin and thiamine were extracted by homogenizing samples in ethanol solvent through a MagNALyser (Roche, Vilvoorde, Belgium, 1 min, 7000 rpm), then centrifuged for 20 min at 14,000× *g*, 4 °C. The supernatant was taken and filtered (Acrodisc GHP filter, 0.45 μm 13 mm). Then, the solution was analyzed by using HPLC (Shimadzu SIL10-ADvp, reverse-phased, at 4 °C), where the target compounds were separated on a reverse-phase (C18) column (HPLC, methanol/water as a mobile phase and fluorescence as a detector) [62].

#### 4.6.1. Total Antioxidant Capacity (FRAP)

Total antioxidant capacity was determined by using the ferric-reducing antioxidant power (FRAP) method. The extraction of samples was performed by using 80% ethanol; then, the extracts were centrifuged for 20 min at 4 °C and 14,000× *g*. The FRAP reagent was prepared by adding FeCl_3_ (20 mM) to the acetate buffer (0.25 M). Thereafter, the FRAP reagent (approximately 0.25 mL) was mixed with 0.1 mL of extracts, and the reading was taken at 593 nm, as previously outlined in [65]. The values were expressed as µmol trolox/g FW.

#### 4.6.2. Amino Acid Analysis

For amino acid analysis, the method described in [66] was used, in which 100 mg of each plant was homogenized in 5 mL of 80% ethanol at 5000 rpm for 1 min. After centrifugation (14,000× *g* for 25 min), the supernatant was resuspended in 5 mL of chloroform. Thereafter, 1 mL of H_2_O was used for the residue extraction. The supernatant and pellet were resuspended in chloroform and centrifuged at 8000× *g* for 10 min. A total of 15 amino acids (0.05 µmoles mL^−1^ for each one) were used as reference standards for determination of the retention time of each amino acid. An internal standard α-aminobutyric was also used for amino acid detection. Then, the extracts were centrifuged for 10 min at 20,000× *g* and the aqueous phase was filtered by Millipore micro-filters (0.2-lm pore size). The amino acids were quantified (using a Waters Acquity UPLC TQD device coupled to a BEH amide column, 2.1 mm × 50 mm). The elution (A, 84% ammonium formate, 6% formic acid, and 10% acetonitrile, *v*/*v*, and B, acetonitrile and 2% formic acid, *v*/*v*) resulted in amino acid peak integration. Star Chromatography (version 5.51) software was applied.

### 4.7. Determination of Nitrogen Content and Metabolism

Total nitrogen (N) content was determined by digestion of the sprout samples (0.2 g) in H_2_SO_4_ at 260 °C; the amount of N was detected by using a CN element analyzer (NC-2100, Carlo Erba Instruments, Milan, Italy). For enzyme assays, the samples (100 mg) were extracted with 400 µL of extraction buffer (50 mM HEPES-KOH pH 7.5, 10% (*v*/*v*) glycerol, 0.1% Triton X-100, 10 mM MgCl_2_, 1 mM EDTA, 1 mM benzamidine, 1 mM ε-aminocapronic acid, 1 mM DTT, and 20 µM flavin adenine dinucleotide). The samples were centrifuged at 4 °C 13,000× *g* for 5 min, and the supernatant was used in the reactions. The determination of glutamine synthetase (GS) and glutamine 2-oxoglutarate aminotransferase (GOGAT) was conducted as indicated by the reduction of NADH at A_340_. Glutamate dehydrogenase (GDH) was determined by 2-oxoglutarate-dependent NADH oxidation. Determination of GS activity was performed by monitoring γ-glutamyl hydroxamate at A_340_. Estimation of GOGAT activity was achieved according to glutamine-dependent NADH oxidation at A340. Nitrate reductase (NR) activity was determined by measuring nitrite-dependent NADH oxidation (A_340_) [44,45]. Arginase was determined according to [67], based on the formation of urea from arginine, where the reaction mixture consisted of 1 mM MnCl_2_, 10 mM Tris (pH 9.5), and 125 mM L-arginine (pH 9.5), in addition to the enzyme solution, to make a total volume of 10 mL. Then, incubation was carried out for 30 min at 37 °C. The reaction was started by adding the enzyme and terminated by the addition of 0.1 mL 50% TCA. Protein removal was performed by centrifugation, and the urea content in the supernatant was colorimetrically measured, where one unit was defined as the amount of enzyme producing 1 umol urea per min. The arginase activity was detected as a linear function of both incubation time and concentration under these conditions. Boiled enzyme preparations were used as the control [67]. Total proteins were detected by using Lowery methods [68].

### 4.8. Determination of Antidiabetic Activity

For sample homogenization, a MagNALyser and a phosphate buffer (1 mL, 50 mM, pH 5.2) were used. Then, centrifugation was carried out for 5 min at 4 °C and 14,000× *g*. The α-amylase inhibitory activity of the extracts and fractions was carried out according to a standard method, with minor modification [69]. In a 96-well plate, the reaction mixture containing a 50 μL phosphate buffer (100 mM, pH = 6.8), 10 μL α–amylase (2 U/mL), and 20 μL of varying concentrations of the extracts (0.1, 0.2, 0.3, 0.4, and 0.5 mg/mL) was preincubated at 37 °C for 20 min. Then, 20 μL of 1% soluble starch (100 mM phosphate buffer pH 6.8) was added as a substrate and incubated further at 37 °C for 30 min; 100 μL of the DNS color reagent was then added and boiled for 10 min. The absorbance of the resulting mixture was measured at 540 nm using a multiplate reader. Acarbose at various concentrations (0.1–0.5 mg/mL) was used as a standard. A without-test substance was set up in parallel as a control, and each experiment was performed in triplicate.

### 4.9. Statistical Analyses

Statistical analyses were completed, using an SPSS statistical package (SPSS Inc., Chicago, IL, USA). Replication of each experiment was performed twice. Four replicates were used for all assays and each replicate corresponded to a group of 25 sprouts harvested from a certain tray. One-way analysis of variance (ANOVA) was carried out, where Tukey’s test was used as the post hoc test for the separation of means (*p* < 0.05). Principal component analysis (PCA) was generated by a multi-experimental viewer (TM4 software package, http://mev.tm4.org, accessed on 18 November 2021).

## 5. Conclusions

Based on the above results, it could be concluded that the application of BABA and/or SeNPs could be a useful technique to enhance the growth and photosynthetic activity of sprouts. As a result, the combined treatment had a more pronounced effect on the bioactive primary metabolites (essential amino acids), secondary metabolises (phenolics, GSH, ASC), mineral profiles, and nitrogen metabolism of the investigated sprouts than that of sole treatments. Concomitantly, the antioxidant (FRAP), the anti-diabetic activities (i.e., α-amylase and α-glucosidase inhibition activities) and the glycemic index) of the tested sprouts were more significantly improved by the combined treatment with BABA and SeNPs than by individual treatment. Thus, this study represents the first report that supports the use of the combined treatment of BABA and SeNPs to increase plant growth and bioactive metabolites.

## Figures and Tables

**Figure 1 plants-11-00306-f001:**
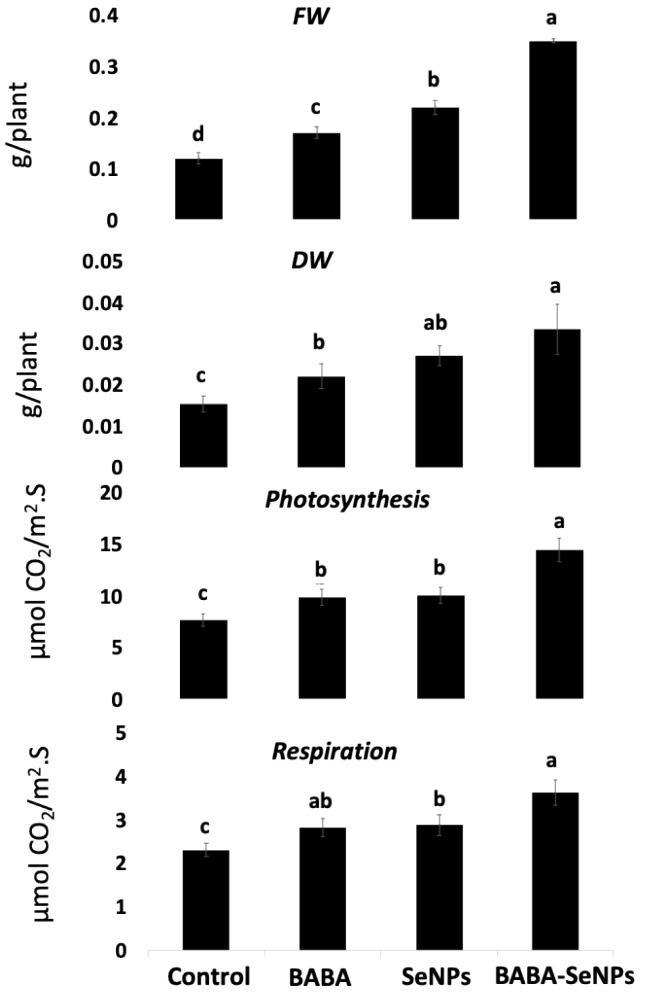
Biomass; fresh weight (FW) (mg g^−1^ FW) and dry weight (DW) (mg g^−1^ FW); photosynthesis (μmol CO_2_ m^−2^ s^−1^); and respiration of control in BABA- and/or SeNPs-treated *M. interexta* sprouts. Data are represented by the means of four replicates ± standard deviations. Different small letter superscripts (a–d) within a row indicate significant differences between control and BABA and/or SeNPs samples. One-way analysis of variance (ANOVA) was performed. Tukey’s test was used as the post hoc test for the separation of means (*p* < 0.05).

**Figure 2 plants-11-00306-f002:**
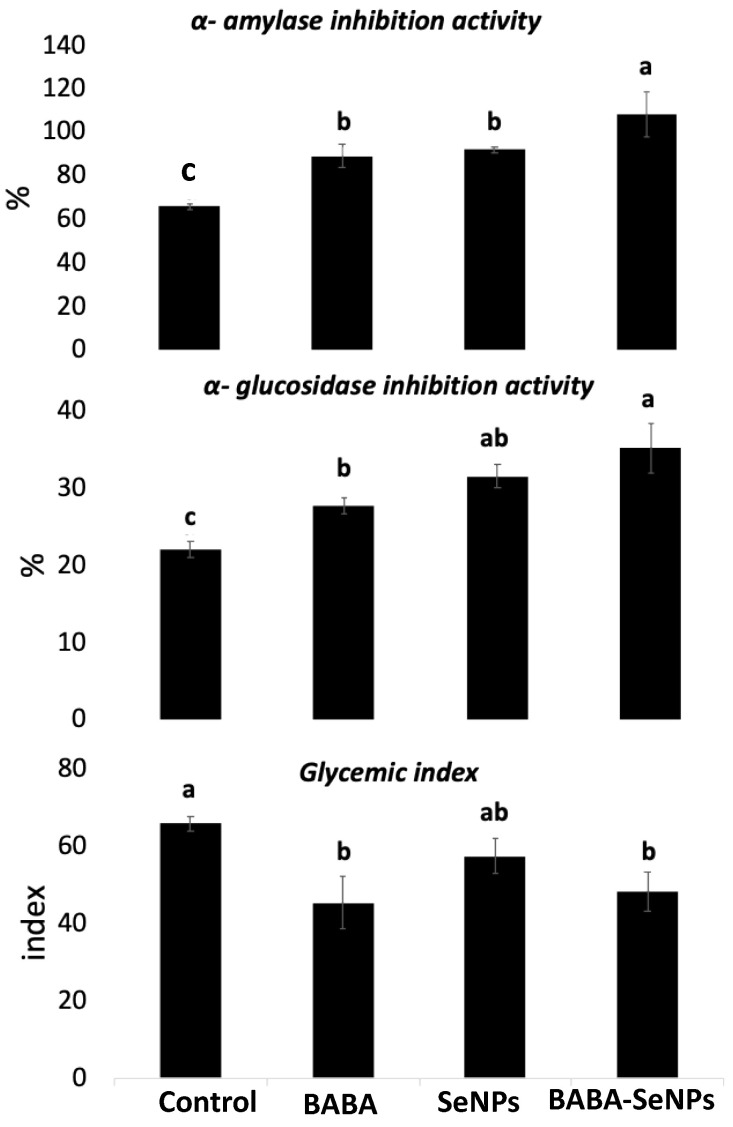
α-amylase and α-glucosidase inhibition activities, and the glycemic index (GI) of control and BABA- and/or SeNPs-treated *M. interexta* sprouts. Data are represented by the means of four replicates ± standard deviations. Different small letters (a–c) within a row indicate significant differences between control and BABA and/or SeNPs samples. One-way analysis of variance (ANOVA) was performed. Tukey’s test was used as the post hoc test for the separation of means (*p* < 0.05).

**Figure 3 plants-11-00306-f003:**
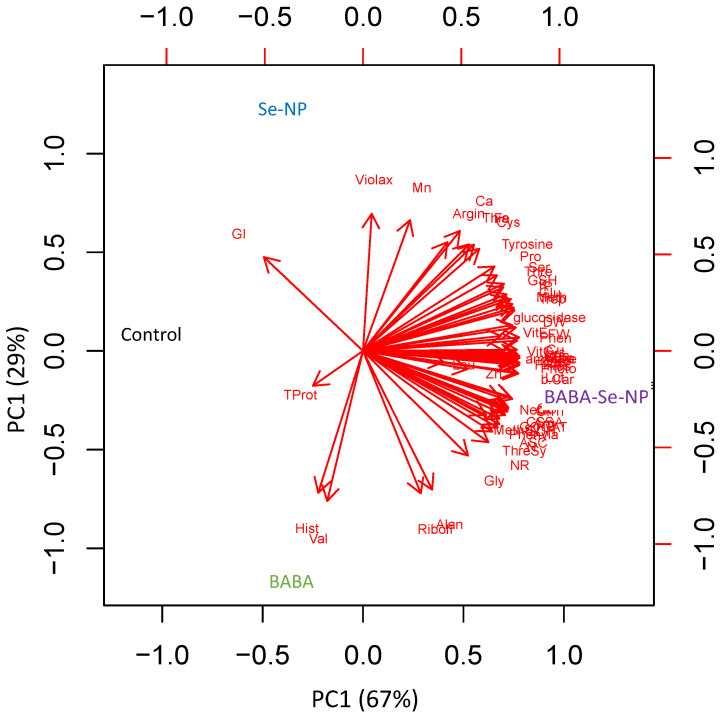
Principal component analysis (PCA) of chemical compositions and biological activities of control and BABA- and/or SeNPs-treated *M. interexta* sprouts.

**Table 1 plants-11-00306-t001:** Pigment content (chlorophyll *a + b*) (mg g^−1^ FW) of control and BABA- and/or Se NPs-treated *M. interexta* sprouts. Data are represented by the means of four replicates ± standard deviations.

	Control	BABA	SeNPs	BABA-SeNPs
Chl a	0.65 ± 0.06 c	0.92 ± 0.02 b	1.05 ± 0.2 b	1.97 ± 0.17 a
Chl b	0.43 ± 0.069 c	0.53 ± 0.08 bc	0.59 ± 0.116 b	1.18 ± 0.19 a
β-Carotene	0.04 ± 0.01 c	0.07 ± 0.004 b	0.07 ± 0.017 b	0.11 ± 0.01 a
Lutein	0.14 ± 0.03 c	0.24 ± 0.02 b	0.23 ± 0.02 b	0.53 ± 0.03 a
Neoxanthin	0.02 ± 0.01 c	0.02 ± 0.003 b	0.01 ± 0.001 c	0.05 ± 0.007 a
Violaxanthin	0.05 ± 0.01 c	0.04 ± 0.003 b	0.07 ± 0.009 a	0.05 ± 0.001 b

Different small letters (a–c) within a row indicate significant differences between control and BABA- and/or SeNPs-samples. One-way analysis of variance (ANOVA) was performed. Tukey’s test was used as the post hoc test for the separation of means (*p* < 0.05).

**Table 2 plants-11-00306-t002:** Mineral elements (mg g^−1^ FW) and vitamins (mg g^−1^ FW) of control and BABA- and/or Se NPs-treated *M. interexta* sprouts. Data are represented by the means of four replicates ± standard deviations.

Parameters	Control	BABA	SeNPs	BABA-SeNPs
**Elements**				
Ca	17.57 ± 2.3 b	15.79 ± 3.5 b	27.79 ± 6.7 a	25.17 ± 0.47 a
Cu	2.26 ± 0.71 b	2.57 ± 1.07 b	2.87 ± 0.28 b	4.38 ± 1.1 a
Fe	3.99 ± 0.23 b	3.15 ± 0.78 b	5.48 ± 1.02 a	5.76 ± 0.44 a
Zn	22.62 ± 2.0 b	36.62 ± 3.3 a	35.88 ± 3.2 a	35.72 ± 3.2 a
Mn	0.25 ± 0.03 a	0.13 ± 0.1 b	0.28 ± 0.13 a	0.27 ± 0.1 a
K	15.60 ± 1.3 c	11.95 ± 3 c	40.60 ± 3.6 b	67.29 ± 6 a
P	5.81 ± 0.6 c	6.48 ± 0.5 c	10.44 ± 0.8 b	13.56 ± 1.1 a
**Vitamins**				
Vit C	7.81 ± 1.3 b	7.31 ± 1.2 b	8.15 ± 2.4 b	13.92 ± 0.7 a
Vit E	47.47 ± 1.2 b	44.57 ± 1.6 cb	48.47 ± 4.4 b	61.92 ± 3.9 a
Thiamin	0.10 ± 0 b	0.07 ± 0 b	0.13 ± 0.02 a	0.14 ± 0.06 a
Riboflavin	0.35 ± 0.3 b	0.51 ± 0.75 a	0.24 ± 0.47 b	0.49 ± 0.96 a

Different small letters (a–c) within a row indicate significant differences between control and BABA and/or Se NPs samples. One-way analysis of variance (ANOVA) was performed. Tukey’s test was used as the post hoc test for the separation of means (*p* < 0.05).

**Table 3 plants-11-00306-t003:** Amino acids (µg g^−1^ FW) of control and BABA- and/or SeNPs-treated *M. interexta* sprouts. Data are represented by the means of four replicates ± standard deviations.

Amino Acids	Control	BABA	SeNPs	BABA-SeNPs
Asparagine	1.53 ± 0.1 b	1.71 ± 0.06 b	1.76 ± 0.02 b	2.17 ± 0.01 a
Glutamine	1.89 ± 0.19 c	2.15 ± 0.25 c	3.39 ± 0.08 b	4.53 ± 0.12 a
Serine	1.18 ± 0.07 c	1.36 ± 0.13 ab	2.31 ± 0.13 b	2.66 ± 0.3 a
Glycine	1.40 ± 0.01 c	1.69 ± 0.07 b	1.16 ± 0.1 c	2.01 ± 0.01 a
Arginine	0.30 ± 0.05 c	0.38 ± 0.08 c	0.77 ± 0.05 a	0.57 ± 0.08 b
Alanine	0.54 ± 0.03 b	0.62 ± 0.03 a	0.51 ± 0 b	0.61 ± 0.02 a
Proline	0.93 ± 0.01 c	1.25 ± 0.03 b	2.54 ± 0.06 a	2.71 ± 0.18 a
Histidine	0.75 ± 0.05 b	0.90 ± 0.05 a	0.67 ± 0.04 b	0.72 ± 0.09 b
Valine	0.76 ± 0.15 b	0.91 ± 0.2 a	0.61 ± 0.09 b	0.73 ± 0.11 b
Methionine	0.66 ± 0.09 c	0.75 ± 0.05 c	0.97 ± 0.01 b	1.17 ± 0.1 a
Cystine	0.99 ± 0.14 b	0.79 ± 0.15 b	1.47 ± 0.08 a	1.56 ± 0.04 a
Ornithine	1.17 ± 0.18 c	2.10 ± 0.21 b	1.72 ± 0.04 b	3.06 ± 0.1 a
Leucine	0.98 ± 0.06 a	0.86 ± 0.18 a	0.86 ± 0.07 a	1.07 ± 0.12 a
Phenylalanine	1.42 ± 0.22 b	1.87 ± 0.23 a	1.65 ± 0.11 b	2.04 ± 0.11 a
Tyrosine	0.31 ± 0.04 a	0.30 ± 0 a	0.42 ± 0.01 ab	0.45 ± 0.01 ab
Lysine	0.70 ± 0.02 b	0.88 ± 0.02 b	1.02 ± 0.03 b	1.91 ± 0.03 a
Threonine	1.18 ± 0.05 b	1.32 ± 0.03 b	1.68 ± 0.08 a	1.78 ± 0.09 a
Treptophane	0.72 ± 0.08 b	0.83 ± 0.1 b	1.06 ± 0.02 ab	1.28 ± 0.04 a

Different small letters (a–c) within a row indicate significant differences between control and BABA and/or SeNPs samples. One-way analysis of variance (ANOVA) was performed. Tukey’s test was used as the post hoc test for the separation of means (*p* < 0.05).

**Table 4 plants-11-00306-t004:** Nitrogen (g 100 g^−1^ FW), protein content (g 100 g^−1^ FW), and nitrogen-related enzymes (umol mg^−1^ protein. min) of control and BABA- and/or SeNPs-treated *M. interexta* sprouts. Data are represented by the means of four replicates ± standard deviations.

	Control	BABA	SeNPs	BABA-SeNPs
Nitrogen	23.3 ± 0.8 b	35.7 ± 0.5 a	28.1 ± 1.2 b	41.2 ± 0.8 a
Total Protein	169.5 ± 1.9 a	118.0 ± 3.1 d	99.6 ± 2.2 c	136 ± 2.8 b
Nitrate reductase	45.2 ± 0.03 c	86.1 ± 5.4 b	43.1 ± 2.2 c	118 ± 11 a
GDH	4.14 ± 0.2 c	6.99 ± 0.48 b	4.9 ± 0.21 c	10 ± 0.48 a
GOGAT	7.8 ± 0.28 d	14.35 ± 0.4 b	10.3± 0.2 c	21 ± 1.8 a
GS	16.12 ± 0.9 d	26.10 ± 0.4 c	23.0 ± 1 b	32 ± 0.8 a
Cyst syn ser acetyltransferase	6.7 ± 0.28 d	11.05 ± 0.0 b	9.0 ± 0.4 c	14.2 ± 0.38 a
Arginase	4.01 ± 0.02 d	7.7 ± 0.46 b	5.9 ± 0.2 cd	10.7 ± 0.9 a
Threonine synthase	1.0 ± 0.02 c	1.70 ± 0.1 b	0.9 ± 0.04 c	2.6 ± 0.17 a
Methionine synthase	2.0 ± 0.01 c	4.30 ± 0.05 a	3.40 ± 0.1 b	4.4 ± 0.2 a

Different small letters (a–d) within a row indicate significant differences between control and BABA and/or Se NPs-samples. One-way analysis of variance (ANOVA) was performed. Tukey’s test was used as the post hoc test for the separation of means (*p* < 0.05).

**Table 5 plants-11-00306-t005:** Flavonoids (mg g^−1^ FW), phenolic acids (mg g^−1^ FW), antioxidant capacity (FRAP) (μmol trolox g ^−1^ FW), GSH (mg g^−1^ FW), and ASC (mg g^−1^ FW) of control and BABA- and/or SeNPs-treated *M. interexta* sprouts. Data are represented by the means of four replicates ± standard deviations.

	Control	BABA	SeNPs	BABA-SeNPs
FRAP	11.9 ± 1.19 c	18.3 ± 2.5 b	18.8 ± 3.6 b	24.0 ± 5.6 a
Phenolics	3.54 ± 0.01 c	5.7 ± 0.02 b	6.7 ± 0.04 b	8.9 ± 0.04 a
Flavonoids	0.58 ± 0.01 c	0.81 ± 0.01 b	0.89 ± 0 b	1.47 ± 0.02 a
Reduced GSH	0.85 ± 0.11 b	1.03 ± 0.3 b	1.1 ± 0.24 a	1.56 ± 0.19 a
Reduced ASC	4.22 ± 0.47 b	7.19 ± 0.69 a	5.4 ± 0.56 b	8.56 ± 0.38 a

Different small letters (a–c) within a row indicate significant differences between control and BABA and/or SeNPs samples. One-way analysis of variance (ANOVA) was performed. Tukey’s test was used as the post hoc test for the separation of means (*p* < 0.05).

## Data Availability

Data is contained within the article.

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
