# Peer review of "Innovating the Synergistic Assets of β-Amino Butyric Acid (BABA) and Selenium Nanoparticles (SeNPs) in Improving the Growth, Nitrogen Metabolism, Biological Activities, and Nutritive Value of Medicago interexta Sprouts"

_plants, 2022, doi:10.3390/plants11030306_

Round 1
Reviewer 1 Report
In general the manuscript is well written and fairly easy to follow. My comments and questions are:
- For selenium treated sprouts, is there any potential concern with causing selenium toxicity towards human who consume the sprouts, since we only require a very low RDA of the element (perhaps 0.005 mg/kg body weight per day)?
- Methods: Contradiction: priming was done with Se-NPs (p. 3), not for the application of BABA as stated p. 11.
- How does Se-NP ppm concentration translate to the molar concentration of Se?
- FRAP assay: 14000 rpm should be expressed as x g.
- How were extracts for enzyme assays prepared?
- What assay was used for arginase determination?
- First paragraph of Methods says experiments were repeated at least twice, last paragraph of methods says they were done two times only. Which is correct?
- Table 3: Ornithine of 2.1 ± 0.21 and 3.06 ± 0.1 are listed as being statistically not different: I don't buy that.
- Final paragraph of results: With reference to Fig. 2, I don't see the eg., 80 % increases in enzyme inhibition indicated in the text. Are the values stated in the text correct?
- Conclusions: the results don't show anything about maintaining cell ultrastructure.
- For the mineral analysis, why aren't Se levels reported? These would be of great interest as relates to the effectiveness of the Se-NP treatments and potential toxicity.
- The Helali 2010 reference is not listed in the citations section.
- Arginase activity was apparently measured and is shown in Table 4, but its significance is never discussed. For example, it might correlate with the higher ornithine levels in the BABA-Se treated seedlings.
- English modifications: p. 2, "One of the inducers is the plant activators a nonprotein amino acid..." needs to be repharased; p 8, arginine misspelled in text and table; p. 9, "...treated with the combined effect..." should be changed to "...treated with the combination of..."; p. 12, use of MI as an abbreviation?; p. 13 "...induces a board of mechanisms..." needs to be restated.
- Amino acid content generally increases with the single or double treaments, yet total protein does not. Does the increased pool of amino acids exist with a higher relative content of free amino acids?
Author Response
In general the manuscript is well written and fairly easy to follow. My comments and questions are:
- For selenium treated sprouts, is there any potential concern with causing selenium toxicity towards human who consume the sprouts, since we only require a very low RDA of the element (perhaps 0.005 mg/kg body weight per day)?
Response: Recently, it has been shown that Se NPs have a more enhancing effect on plants with a low toxicity when compared with the bulk form, according to the following references:
-Li, X.; Xu, H.; Chen, Z.S.; Chen, G. Biosynthesis of nanoparticles by microorganisms and their applications. J. Nanomater. 2011.
-Gupta, M.; Gupta, S. An Overview of Selenium Uptake, Metabolism, and Toxicity in Plants. Front. Plant Sci.2017.
So, the impact of selenium treated sprouts on human health might be dependent on selenium size.
- Methods: Contradiction: priming was done with Se-NPs (p. 3), not for the application of BABA as stated p. 11.
Response: we corrected it in p. 11
- How does Se-NP ppm concentration translate to the molar concentration of Se?
Response: Here we expressed the concentration of Se-NP as PPM (mg/L). Molar concentration is used mostly for homogeneous solutions, thus I think that g/l concentration is meaningful for nanoparticle suspensions, while mol/l concentration should be used for homogeneous solutions
- FRAP assay: 14000 rpm should be expressed as x g.
Response: thanks, done
- How were extracts for enzyme assays prepared?
Response: the method for enzyme assays was included in the text
- What assay was used for arginase determination?
Response: the assay used was based on the formation of urea from arginine, and we described it in the method
- First paragraph of Methods says experiments were repeated at least twice, last paragraph of methods says they were done two times only. Which is correct?
Response: we repeated the experiment exactly 2 times, and we edited this in the text
- Table 3: Ornithine of 2.1 ± 0.21 and 3.06 ± 0.1 are listed as being statistically not different: I don't buy that.
Response: thanks, we corrected this in the table to be statistically different
- Final paragraph of results: With reference to Fig. 2, I don't see the eg., 80 % increases in enzyme inhibition indicated in the text. Are the values stated in the text correct?
Response: thanks, we corrected this in the text to be 40% increase
- Conclusions: the results don't show anything about maintaining cell ultrastructure.
Response: we deleted it
- For the mineral analysis, why aren't Se levels reported? These would be of great interest as relates to the effectiveness of the Se-NP treatments and potential toxicity.
Response: thanks, but (Se) standard was not available during our measurements
- The Helali 2010 reference is not listed in the citations section.
Response: thanks, we included it in the reference list
- Arginase activity was apparently measured and is shown in Table 4, but its significance is never discussed. For example, it might correlate with the higher ornithine levels in the BABA-Se treated seedlings.
Response: we added more details about arginase activity, and its impact on the glutamine synthetase/glutamine oxoglutarate aminotransferase (GS/GOGAT) cycle
- English modifications: p. 2, "One of the inducers is the plant activators a nonprotein amino acid..." needs to be repharased; p 8, arginine misspelled in text and table; p. 9, "...treated with the combined effect..." should be changed to "...treated with the combination of..."; p. 12, use of MI as an abbreviation?; p. 13 "...induces a board of mechanisms..." needs to be restated.
Response: thanks, we corrected and edited all these sentences
- Amino acid content generally increases with the single or double treaments, yet total protein does not. Does the increased pool of amino acids exist with a higher relative content of free amino acids?
Response: Yes, actually the percentage of free amino acids was higher than that of protein amino acids
Reviewer 2 Report
The paper titled „Innovating the synergistic assets of β-amino butyric acid (BABA) and selenium nanoparticles (SeNPs) on improving the growth, nitrogen metabolism, biological activities, and nutritive value of Medicago interexta sprouts” is interesting and well prepared. However, there are some questions/comments that should be answered before its acceptance for publication.
Introduction:
- Authors stated that selenium nanoparticles play a significant role in activating the defence system of plants. In my opinion some detail information should be mentioned how selenium influence plant growth and its physiological parameters as well as what is the concentration of selenium (in particle and nanoparticle form) that is toxic for the plant? What are the safe concentration of Se NPs for plants (that do not bring toxic effect)?
Methods:
- How many samples were analyzed in total?I think that the pattern of the carried out experiments would facilitate the understanding of how many different variants of the research were conducted.
- How the concentrations of BABA and Se NPs were selected?
- Were whole sprouts analyzed or only their parts?There is no information on this anywhere in the text.Please complete the description.This is of particular importance for the photosynthetic rate and pigment analysis determination.
- How were the plants selected for analysis?
- Do the authors have sample photos of the sprouts after cultivation in different variants of the experiment?
Results
- In the descriptions of figures and tables it is stated that the results include the mean of at least 3 replicates + standard deviation.Exactly how many repetitions were there, were plants collected for the determination at least three sprouts from one variant of the experiment, was each variant repeated at least three times and at least three sprouts were collected from each variant?Please specify.
- There are some editorial errors in the text - please correct
- What role do thiamin and riboflavin play in plant tissue that the authors decided to determine them?
Discussion
- The authors report that Se NPs improve plant growth.Can Se NPs also be toxic to plants?What it depends on?
- How do Se NPs and BABA separately affect plants?Can the authors unequivocally state whether NPs Se is photosynthetic positive?
- Please correct editorial errors - some words are underlined but should not be.Please standardize the wording chl a chl b (upper or lower case letters) - throughout the text and in the Table.
- The authors found that BABA increased Zn concentration in sprouts, and SeNPs increased the concentration of K and P. What is the mechanism of action of the compounds used that contributed to the growth of these elements?Please explain in detail.
- Can the increase in the concentration of flavonoids and phenols be related to the toxic effect of BABA and / or Se NPs? Please explain.
- The second part of the chapter on "improved antidiabetic activity ..." is speculative.Please correct it.
Conclusions
There are no conclusions in which the authors clearly refer to research hypotheses (especially to the influence of BABA and/or Se NPs on the nutritional and pharmacological value of M. interexta) ??

Author Response
The paper titled „Innovating the synergistic assets of β-amino butyric acid (BABA) and selenium nanoparticles (SeNPs) on improving the growth, nitrogen metabolism, biological activities, and nutritive value of Medicago interexta sprouts” is interesting and well prepared. However, there are some questions/comments that should be answered before its acceptance for publication.
Introduction:
- Authors stated that selenium nanoparticles play a significant role in activating the defence system of plants. In my opinion some detail information should be mentioned how selenium influence plant growth and its physiological parameters as well as what is the concentration of selenium (in particle and nanoparticle form) that is toxic for the plant? What are the safe concentration of Se NPs for plants (that do not bring toxic effect)?
Response: we added more details in the introduction
Methods:
- How many samples were analyzed in total? I think that the pattern of the carried out experiments would facilitate the understanding of how many different variants of the research were conducted.
Response: Thanks all details are added.
- How the concentrations of BABA and Se NPs were selected?
Response: Based on a preliminary experiment, we selected the most effective concentrations of BABA and Se NPs
- Were whole sprouts analyzed or only their parts? There is no information on this anywhere in the text. Please complete the description. This is of particular importance for the photosynthetic rate and pigment analysis determination.
Response: we used the leaves and stems of sprouts for different analyses, and we have demonstrated this in the experimental setup
- How were the plants selected for analysis?
Response: we selected the healthiest plants with the best growth, as indicated from the photosynthetic growth parameters.
- Do the authors have sample photos of the sprouts after cultivation in different variants of the experiment?
Response: yes, to be sure about the reproducibility of the two experiments, growth and physiological analyses were performed for each experiment and the results were very similar
Results
- In the descriptions of figures and tables it is stated that the results include the mean of at least 3 replicates + standard deviation. Exactly how many repetitions were there, were plants collected for the determination at least three sprouts from one variant of the experiment, was each variant repeated at least three times and at least three sprouts were collected from each variant? Please specify.
Response: Thanks, details are added to methodology and results
Each experiment was replicated two times, and for all assays, 4 biological replicates (two biological replicates from each experiment) were used, thus in total 16 samples were analyzed per each measurement. Moreover, each replicate corresponded to a group of 25 sprouts harvested from a certain tray.
- There are some editorial errors in the text - please correct
Response: we have carefully checked the whole manuscript for all errors, and corrected them
- What role do thiamin and riboflavin play in plant tissue that the authors decided to determine them?
Response: previous studies have demonstrated that thiamin and riboflavin play an important role as coenzymes for many metabolic enzymes, which are involved in photosynthesis and respiration, So, we wanted to determine them in our study.
Discussion
- The authors report that Se NPs improve plant growth. Can Se NPs also be toxic to plants? What it depends on?
Response: Yes, Se- NPs toxicity occurs in plants when optimum concentration of Se-NPs is exceeded. However, at certain concentration, Se NPs have a more enhancing effect on plants with a low toxicity when compared with the bulk form. Moreover, So, the impact of Se on plant growth might be dependent on its size
-Li, X.; Xu, H.; Chen, Z.S.; Chen, G. Biosynthesis of nanoparticles by microorganisms and their applications. J. Nanomater. 2011.
-Gupta, M.; Gupta, S. An Overview of Selenium Uptake, Metabolism, and Toxicity in Plants. Front. Plant Sci.2017.
- How do Se NPs and BABA separately affect plants? Can the authors unequivocally state whether NPs Se is photosynthetic positive?
Response: Thanks we added more details of positive growth promoting effect of Se and BABA to the discussion. Indeed, previous studies have shown that Se NPs could enhance the photosynthetic efficiency of some plants such as tomato. Such positive effect could be reflected on increasing the pigment contents, as reported also in our study. This might be due to the small size of nano-particles, so they can easily move through the plant parts.
BABA has been supposed to enhance nitrogen metabolism, which consequently provide precursors needed for amino acids and protein biosynthesis, and consequently increase photosynthesis, growth rate and biomass accumulation.
- Please correct editorial errors - some words are underlined but should not be. Please standardize the wording chl a chl b (upper or lower case letters) - throughout the text and in the Table.
Response: we revised and corrected all words throughout the text
- The authors found that BABA increased Zn concentration in sprouts, and SeNPs increased the concentration of K and P. What is the mechanism of action of the compounds used that contributed to the growth of these elements? Please explain in detail.
Response: the BABA-induced increases in N availability to could be due increased root growth which in turn, triggers the nutrient uptake by plants. BABA might be able to increase mineral transporters. Moreover, improved nitrogen nutrition by BABA could enhance root uptake, root-to-shoot translocation and remobilization of some mineral such as Zn. Consequentially, increased level of Zn is needed for biosynthesis and structural and functional integrity of proteins and amino acid metabolism. We added these details to the discussion part.
Regarding SeNPs effect on mineral uptake, the study of Pandey and Gupta (2018) indicated that exposure to Se significantly upregulated expressions of the phosphate transporter (PHT) and the potassium channel protein (KCP) and potassium transporter protein (KTP). In agreement with our results, Se application was found to enhance the mineral content (e.g. Zn, Mn, Cu, Ca, Mg, Na, and K) of alfalfa and radish, Unlike our results, a previous study did not significant changes in the concentration of minerals detected in lettuce.
Can the increase in the concentration of flavonoids and phenols be related to the toxic effect of BABA and / or Se NPs? Please explain.
Response: It has been previously reported that BABA and / or Se NPs could stimulate the phenylalanine ammonia-lyas, which is a key enzyme in phenylpropanoid pathway, as being responsible for biosynthesis of phenolic compounds, hence increasing the phenolic content was not related to the toxic effect of BABA and / or Se NPs. In addition, the induced photosynthetic activity under BABA and / or Se could significantly increase the carbon input necessary for biosynthesis of different classes of secondary metabolites such as phenolic compounds
- The second part of the chapter on "improved antidiabetic activity ..." is speculative. Please correct it.
Response: we corrected it
Conclusions
There are no conclusions in which the authors clearly refer to research hypotheses (especially to the influence of BABA and/or Se NPs on the nutritional and pharmacological value of M. interexta) ??
Response: we added the conclusion part
Reviewer 3 Report
Due to low scientific level and low technical quality of manuscript preparation, this paper cannot be accepted. My individual comments are listed below.
The language must be corrected by a native speaker in English.
Page 1:
Capital first letters in the title.
e-mail addresses and authors’ initials.
Ppm is not any SI unit.
It should be “phenolics”.
Flavonoids belong to phenolics.
It should be “glycemic index”.
Page2:
References must be cited using […}.
It should be “β-amino butyric acid”.
Flavonoids belong to phenolics.
It should be “nanoparticles (NPs)”.
Page 3:
Supplier of Giza 72?
Ppm is not any SI unit.
Full name of PAR.
It should be “American Elements”.
It should be “gray to black”.
It should be “Photosynthesis was determined …”.
What was the sample/
Page 4:
The extraction of pigment and phenolic analysis should be described.
How the sample for HPLC analysis was prepared?
It should be “chlorophyll”.
In my opinion, it was C18 not Si60 column. The used gradient must be described.
The centrifugation must be characyerized by “x g” instead of “rpm”.
How FRAP was expressed?
Extraction for FRAP should be described.
517 nm is used for DPPH assay not FRAP.
Mobile phase for AA HPLC analysis?
It should be “The amino acids were detected at …”.
Page 5:
Reference for enzyme activity determination?
Determination of the amylase determination should be much better described.
It should be “one-way analysis of variance”.
Figure 1 Table 1 – The highest value must be marked with “a”, lower with “b”, etc.
Table 2 It should be “Ornithine”.
Table 2, 3 – The units must be reported.
Table 3 “Trep”?
Page 7 – The vitamin E activity show tocopherols and tocotrienol. Which compounds were determined using an HPLC?
Typical HPLC of phenolics, vitamins, and amino acids must be reported.
Table 4 – The units must be added. Some description “letters) of statistical analysis are missing. A big part of the title must be moved to the footnote.
Table 5 – The content of individual phenolic acids and flavonoids determined using HPLC must be reported.
Page 9 - Flavonoids belong to flavonoids.
FRAP is a method not antioxidant!
Page 15:
Author Contribution and Funding should be reported
References must be in order of citation. The MDPI style (template) must be used. The Latin names must be written with italic.
Author Response
The language must be corrected by a native speaker in English.
Response: English language were critically checked and revised
Page 1:
Capital first letters in the title.
Response: done
e-mail addresses and authors’ initials.
Response: done
Ppm is not any SI unit.
Response: we corrected this
It should be “phenolics”.
Response: corrected
Flavonoids belong to phenolics.
Response: corrected
It should be “glycemic index”.
Response: corrected
Page2:
References must be cited using […}.
Response: done
It should be “β-amino butyric acid”.
Response: done
Flavonoids belong to phenolics.
Response: we edited it
It should be “nanoparticles (NPs)”.
Response: we edited it
Page 3:
Supplier of Giza 72?
Response: we added it
Ppm is not any SI unit.
Response: corrected
Full name of PAR.
Response: we added it
It should be “American Elements”.
Response: done
It should be “gray to black”.
Response: done
It should be “Photosynthesis was determined …”.
Response: done
What was the sample/
Response: it was sprout sample, and we clarified this in the method
Page 4:
The extraction of pigment and phenolic analysis should be described.
Response: we added the method of extraction of phenolics, but the extraction of pigment was already described
How the sample for HPLC analysis was prepared?
Response: we preparation of samples in the method
It should be “chlorophyll”.
Response: done
In my opinion, it was C18 not Si60 column. The used gradient must be described.
Response: done
The centrifugation must be characyerized by “x g” instead of “rpm”.
Response: done
How FRAP was expressed?
Response: it was expressed as µmol trolox/g FW, and we added this in the method.
Extraction for FRAP should be described.
Response: it was already described in the method.
517 nm is used for DPPH assay not FRAP.
Response: we corrected it (593 nm)
Mobile phase for AA HPLC analysis?
Response: The mobile phase A and B are 10% 200 mmol/L ammonium formate-water solution and 10% 200 mmol/L ammonium formate-acetonitrile solution, respectively, and we added it in the method
It should be “The amino acids were detected at …”.
Response: done
Page 5:
Reference for enzyme activity determination?
Response: we added the required references
Determination of the amylase determination should be much better described.
Response: we added more details
It should be “one-way analysis of variance”.
Response: done
Figure 1 Table 1 – The highest value must be marked with “a”, lower with “b”, etc.
Response: done as required
Table 2 It should be “Ornithine”.
Response: done
Table 2, 3 – The units must be reported.
Response: The units were provided
Table 3 “Trep”?
Response: we corrected it (Treptophane)
Page 7 – The vitamin E activity show tocopherols and tocotrienol. Which compounds were determined using an HPLC?
Response: tocopherols was determined
Typical HPLC of phenolics, vitamins, and amino acids must be reported.
Response: we already provided the HPLC method
Table 4 – The units must be added. Some description “letters) of statistical analysis are missing. A big part of the title must be moved to the footnote.
Response: we added the units and the missing letters
Table 5 – The content of individual phenolic acids and flavonoids determined using HPLC must be reported.
Response: Actually we did not determine the individual phenolic acids and flavonoids, but we determined their totals, and we edited this in the method section.
Page 9 - Flavonoids belong to flavonoids.
Response: we edited it
FRAP is a method not antioxidant!
Response: we corrected it
Page 15:
Author Contribution and Funding should be reported
Response: we added it
References must be in order of citation. The MDPI style (template) must be used. The Latin names must be written with italic.
Response: done
Round 2
Reviewer 2 Report
The manuscript was correxted on accordance to my sugestion. It can be published on present form.
Author Response
Thank you very much
Reviewer 3 Report
The authors ignored many of my comments. Language is still unacceptable. I don't believe the publication was improved by a native speaker. The changes have not been marked in color. The citations in the text are not Plants style. The differences between mean values are marked in wrong way. The style of many references is wrong. How the authors could determine tryptophane after 6 M HCl hydrolysis? Determination and calculation of vit. E is not described. The used gradient for HPLC analyses is not reported.
Author Response
The authors ignored many of my comments. Language is still unacceptable. I don't believe the publication was improved by a native speaker.
Response: Thanks We carefully checked the manuscript to avoid any typos and to improve the presentation quality. We belive our manuscript is well written, and this is also supported by the other first and second reviewers who stated that “In general the manuscript is well written and fairly easy to follow” & is interesting and well prepared”, respectively
The changes have not been marked in color.
Response: Thanks All changes made in red color
The citations in the text are not Plants style. The differences between mean values are marked in wrong way. The style of many references is wrong.
Response: Thanks, we edited all citations to Plants style
How the authors could determine tryptophane after 6 M HCl hydrolysis?
Response: thanks, corrected and more details are added
Determination and calculation of vit. E is not described.
Response: Thanks all details on vitamins methods are added
The used gradient for HPLC analyses is not reported.
Response: Thanks, added
Round 3
Reviewer 3 Report
The authors corrected this paper properly taken under considerations all my comments. Therefore, I can accept it now.